# Assembly of Biologically Functional Structures by Nucleic Acid Templating: Implementation of a Strategy to Overcome Inhibition by Template Excess

**DOI:** 10.3390/molecules27206831

**Published:** 2022-10-12

**Authors:** Matthew M. Lawler, James T. Kurnick, Leah Fagundes St. Pierre, Estelle E. Newton, Lenora B. Rose, Ian S. Dunn

**Affiliations:** 1TriBiotica LLC, 100 Cummings Center, Suite 424-J, Beverly, MA 01915, USA; 2Department of Pathology, Massachusetts General Hospital, Boston, MA 02114, USA

**Keywords:** nucleic acids, templated assembly, chemical modification, molecular proximity, antibody epitope, template titration

## Abstract

Delivery of therapeutic molecules to pathogenic cells is often hampered by unintended toxicity to normal cells. In principle, this problem can be circumvented if the therapeutic effector molecule is split into two inactive components, and only assembled on or within the target cell itself. Such an in situ process can be realized by exploiting target-specific molecules as templates to direct proximity-enhanced assembly. Modified nucleic acids carrying inert precursor fragments can be designed to co-hybridize on a target-specific template nucleic acid, such that the enforced proximity accelerates assembly of a functional molecule for antibody recognition. We demonstrate the in vitro feasibility of this adaptation of nucleic acid-templated synthesis (NATS) using oligonucleotides bearing modified peptides (“haplomers”), for templated assembly of a mimotope recognized by the therapeutic antibody trastuzumab. Enforced proximity promotes mimotope assembly via traceless native chemical ligation. Nevertheless, titration of participating haplomers through template excess is a potential limitation of trimolecular NATS. In order to overcome this problem, we devised a strategy where haplomer hybridization can only occur in the presence of target, without being subject to titration effects. This generalizable NATS modification may find future applications in enabling directed targeting of pathological cells.

## 1. Introduction

Conventional cancer chemotherapeutic agents are primarily directed towards rapidly dividing cells, resulting in toxicity towards certain normal cellular bystander populations in addition to the desired tumor targets. Such effects can cause significant morbidity, limiting the efficacy of many cytotoxic therapies. While there have been fundamental advances in the use of immunotherapeutic targeting of tumors through immune checkpoint antibodies [1,2,3] and effector cells with chimeric receptors [4,5,6], there remain significant limitations to these technologies [7,8]. In principle, non-specific bystander toxicity could be greatly reduced if a therapeutic agent is assembled only in situ in (or on) the pathological target cell of interest. By assembling two otherwise inert fragments into a functional therapeutic on a nucleic acid template that is present only on the pathological cells, it may be possible to provide such therapeutic benefit.

Nucleic acid templating for molecular synthesis and reaction rate enhancement has been applied and developed in a variety of circumstances and architectures [9,10,11,12,13,14], and this technology is a potential pathway for allowing in situ assembly of therapeutic molecules. Accordingly, we have exploited nucleic acid templating processes (NATS; nucleic acid-templated synthesis), involving two oligonucleotides each conjugated to mutually interactive moieties. For simplicity, we have termed such oligonucleotide conjugates as “haplomers”, denoting their singular, or incomplete functional status prior to templating and full product assembly (Figure 1B,C). An essential feature of NATS-based molecular templating is the enforcement of molecular proximity between two participating haplomers, such that their effective concentrations are dramatically increased over concentrations in free solution. By extension, successful application ofNATS has two major requirements: each haplomer in isolation should be inert, and the two haplomers should only form an active product in the presence of specific template harbored by target cells of interest.

Of fundamental significance from the earliest stages of molecular evolution, biologically based templating [15,16,17] can be adapted for directed catalysis of specific reactions [18,19,20]. Assembly of a desired product on a unique RNA template strand (mRNA or non-coding) has the potential for exquisite focus upon the target cell of interest, thereby minimizing toxic bystander effects. In addition, nuclease-resistant artificial nucleic acid templates can be displayed on the surfaces of target cells, if such templates are appended with a ligand for a known target-specific surface receptor. 

Targeting of cell-specific transcripts as templates has been proposed for triggering activation of pro-drug molecules [21,22]. While of considerable promise, a limitation of this in situ approach is the generation of sufficient low molecular weight toxic product to destroy the target host cell, if product formation is restricted to an equimolar ratio with the template. Cyclic turnover of template-driven product accumulation could in principle circumvent this problem, where the template would approach a true catalytic role. This has been attained with certain in vitro systems [23,24], but is difficult to establish with an endogenous nucleic acid target. The potency barrier for effective toxicity may be potentially overcome if templated product formation can be functionally amplified through its recognition by immune effectors, including T cells and antibodies. Templated peptide assembly is a promising route towards this goal, wherein intracellular specific peptide products may be directed into the pathway for surface display on major histocompatibility complex (MHC) molecules. Furthermore, directed assembly of peptide-bearing haplomers on the surface of a target pathological cell can enable specific cellular flagging for recognition by specific antibodies. 

In this report we demonstrate the functional utility of NATS through the in vitro templated chemical reconstitution of an epitope mimic (mimotope) recognized by the therapeutic anti-HER2/neu antibody trastuzumab, using traceless native chemical ligation chemistry with appropriately constructed haplomers. At the same time, we address a potentially serious limitation to the general deployment of a templatable binary haplomer system: the template titration problem. An inherent property of templated bimolecular reactions is their requirement for an equimolar ratio between the two templated components (haplomers) and the template itself, for a single cycle of product formation (Appendix A, Architecture 2). In such circumstances, if the starting-point molar levels of the template exist in significant excess above the haplomer levels, then a predictable outcome is a fall in the efficiency of product formation, through template-mediated titration of individual haplomers. While normal mRNA transcripts are often present at low steady-state levels, it has long been known that many cancer-relevant RNAs are highly overexpressed [25,26,27].

In any case, when attempting to use nucleic acid templates present within cells of interest, it may be difficult to estimate either the effective target template level, or the effective haplomer levels achievable in practice. While transcript levels may be determined on a population basis, considerable stochastic variation can still exist between single cells, even for those of the same recent ancestry [28]. Moreover, even if accurate information for levels of a specific transcript is assumed, dynamic secondary structures may perturb the effective accessible concentration for templating. Haplomer levels attainable in a potential therapeutic context will be contingent on both pharmacokinetic/pharmacodynamic factors and delivery strategies, rendering it a challenging prospect for generating a definable ratio of haplomers to target template in situ. Even when successfully delivered to cells of interest, only a fraction of intracellular haplomers may become accessible to target RNAs, by analogy with experience gained from molecular beacons [29].

For these reasons, and the demonstrable inhibitory effects of excessive template concentrations, we investigated a strategy to circumvent the template titration problem, termed “Locked-NATS.” This process is dependent on structurally determined differential hybridizations where only the presence of the designated target template allows mutual haplomer hybridization and proximity-driven formation of product. It can be distinguished from the initial NATS templating process (Appendix A, Architecture 2) by its use of the Architecture 1 configuration (Appendix A) for generating haplomer proximity. For simplicity, we refer to the original Architecture 2 NATS as “linear NATS” to contrast its configuration from Locked-NATS. 

## 2. Results

### 2.1. Haplomer Structures and Design

Templated synthesis as enabled by NATS is dependent on coupling of mutually reactive groups to nucleic sequences brought into proximity by directed hybridizations, which can be mediated by a number of different molecular architectures [9,20,30]. In this work, we define haplomers as bipartite functional units fundamental to NATS technology. Each haplomer is composed of an oligomeric nucleic acid component covalently coupled with an independent molecular entity that is capable of a chemical reaction (or specific non-covalent interaction) with a specific third-party molecule, to yield a desired product. When the third-party itself is similarly coupled with a separate nucleic acid oligomer in an appropriate configuration, directed hybridizations can force the reactive molecules into spatial proximity, with concomitant strong acceleration of reaction kinetics and product formation. These arrangements are depicted in Appendix A, where the independent mutually reactive molecular groups that are coupled to two distinct nucleic acid oligomers are generalized as α and β. Two examples of molecular architectures relevant to this work are depicted in Appendix A. Mutual bimolecular complementarity between two haplomer nucleic acid components (here termed Architecture 1; Appendix A) elicits the desired spatial proximity of α and β conjugated to 5′ and 3′ ends of each nucleic acid, by virtue of normal antiparallel duplex formation. Alternatively, comparable spatial proximity of α and β can be achieved where each nucleic acid segment is complementary to spatially close regions of a common template (Architecture 2; Appendix A). The functionally bipartite nature of haplomer structures can be further dissected (Appendix A) into the chemistries that enable both the linking of defined nucleic acid oligomers with specifically reactive moieties (α and β of Appendix A), and the interaction of haplomer-borne reactive moieties to form a defined product. To maximize the effective functionality of the latter moieties, their chemical reactivities should be restricted towards each other. In a biological context, this is realized when their respective reactivities are fully bioorthogonal, which can be attained by traceless Staudinger chemistry or click chemical reactions [31,32].

### 2.2. NATS Production of an Antibody Mimotope Peptide

The splitting and subsequent functional reconstitution of an antibody epitope was tested as a proof of principle of this immunological application of NATS. To demonstrate NATS efficacy, we thus chose a well-characterized antibody, trastuzumab (Herceptin) used therapeutically for targeting cancer cells overexpressing the HER-2/Neu member of the EGFR family [33]. Although the structural basis of HER-2 recognition by trastuzumab has been defined [34], it was convenient to make use of a short peptide mimotope (QLGPYELWELSH [35], which binds to trastuzumab with high specificity. Since native chemical ligation (NCL) affords a simple approach to traceless peptide ligation, provided a cysteine residue is present [36] and has been demonstrated to be accelerated by appropriate nucleic acid templating [37], we tested the replacement of several residues of this mimotope with cysteines for retention of trastuzumab recognition. One of these modifications where the single serine residue was substituted (QLGPYELWELCH) elicited comparable binding by trastuzumab as achieved with the original mimotope (Appendix A).

To test the efficacy of epitope assembly by NCL, we used this mimotope information for the synthesis of two modified peptides (Figure 1A), where the first (peptide A444) carries a C-terminal phenyl thioester, the second (peptide A423) has an N-terminal cysteine, and both have respective N-terminal or C-terminal azide groups to enable subsequent oligonucleotide conjugation by click chemistry. Even without template-assisted proximity enhancement, co-incubation of these two peptides was expected to allow their chemical ligation in significant yields if concentration and time parameters were suitably adjusted. Both to assess full mimotope assembly and trastuzumab recognition towards the component peptides alone, we used an inhibition assay. Here, either peptide component alone or the co-incubation product was pre-incubated with trastuzumab, prior to ELISA testing for the same antibody binding to intact QLGPYELWELCH peptide. It was found (Appendix A) that neither of the two component peptides in isolation had any significant effect on the specific trastuzumab signal, whereas the co-incubation product markedly reduced the signal, in a dose-dependent manner. Since application of the NATS principle requires that haplomer components show no significant activity in isolation, the trastuzumab mimotope segments for NCL thus fulfilled this requirement. 

### 2.3. Production of Haplomers for Trastuzumab Mimotope Assembly

Enforcement of proximity of hybridizing haplomers can be achieved with either Architecture 1 or 2, as defined (Appendix A). Thus, we used three oligonucleotides (#439, #440, and #441), where #440 and #441 are mutually complementary, and #440 and #439 are complementary to a common template. Oligonucleotides #439 and #441 have 3′ DBCO-modifications, and #440 has a corresponding 5′-DBCO group. The modified trastuzumab peptide segments (A444 and A423; Figure 1A) were conjugated to oligonucleotides by means of Cu-free strained-click reactions [38] between N- or C-terminal peptide azide groups, and the 5′- or 3′- DBCO oligonucleotide modifications. In this manner, the DBCO-modified 5′ end of oligonucleotide #440 was conjugated with the N-terminal azide of peptide A444 to form haplomer **[a]** (Figure 1B,C), and the DBCO-modified 3′ ends of oligonucleotides #441 and #439 were conjugated with the C-terminal azide of peptide A423 to produce haplomers **[b]** and **[c]**, respectively (Figure 1B,C). Details of the click reactions are shown in the Figure 1B subpanel. Isomeric heterogeneity of the resulting products does not affect subsequent haplomer templating and functional reactivity. When the nucleic acid components of haplomers **[a]** and **[b]** mutually hybridize, or the nucleic acid components of haplomers **[a]** and **[c]** hybridize to a common template, in principle NCL reactions can be promoted, with formation of the original mimotope (Figure 1B,C). 

In denaturing acrylamide gels, the peptide-conjugated oligonucleotides showed an altered mobility from free oligonucleotides, with the reaction driven towards completion in conditions of high peptide molar excess (Figure 1D). During the conjugation process, oxidative reactions between excess cysteine-bearing peptide A423 with haplomers **[****b]** and **[****c]** were evident from their apparent slower electrophoretic mobilities relative to the larger A444-oligonucleotide #440 conjugate (Figure 1D). This was confirmed and easily overcome by treatment with the reducing agent Tris-(2-carboxyethyl) phosphine (TCEP). The electrophoretic mobilities of peptide-oligonucleotide conjugates under these conditions showed similar, but not identical mobilities to that expected for normal oligonucleotides of the same number of residues. For example, haplomer [**a]**, composed of a 17-mer oligonucleotide conjugated via NCL to a 10-mer peptide, showed a mobility of ~30 bases relative to the single-stranded DNA size ladder (Figure 1D). Similarly, after TCEP reduction, haplomers **[b]** and **[c]** (16- and 17-mer oligonucleotides, respectively) when reacted with the 6-mer peptide A423 (Figure 1A) showed an apparent mobility of ~23 bases.

### 2.4. NCL-Enabled Templated Peptide Assembly and Mimotope Recognition by Trastuzumab

When haplomers **[****a]** and **[****b]** with mutually complementary nucleic acid components (Figure 2A) were mixed and allowed to react, a product band was observed under denaturing gel electrophoretic conditions (Figure 2A, Lane 4). In contrast, haplomers **[****a]** and **[****c]**, whose nucleic acid components were non-complementary, showed no such product when mixed at the same concentrations and time periods (Figure 2A, Lane 5). Product formation was seen, however, when **[****a]** and **[****c]** were co-incubated with a mutually complementary template, but not a scrambled template (Figure 2A, Lanes 6 and 7, respectively). The apparent electrophoretic mobility for the templated **[****a]** + **[****c]** product (~55 bases relative to the DNA size markers) conformed to the previous observations (Figure 1), but the mobility of the **[****a]** + **[****b]** product was anomalously fast. Since this effect was only seen for self-complementary Architecture 1 products (Figure 1B), it is most likely due to stabilization of the oligonucleotide duplex after NCL product formation, even under the denaturing conditions used. 

The functional assembly of the peptide mimotope was assessed by its binding by trastuzumab in ELISA assays with immobilized template oligo for haplomers **[****a]** and **[****c]**. Following co-incubations for templated in situ NCL reactions, the antibody produced significant signals from the haplomer pair, but nothing from either haplomer in isolation (Figure 2B). 

Under the conditions used, hybridization of haplomer pairs to immobilized template proceeded rapidly, as detected by varying the time of exposure of template to haplomers followed by a wash step, prior to an equalized time for NCL reactivity (Figure 2C). With a standard 10 min hybridization time, NCL reactivity as assessed by trastuzumab ELISA assay was found to be rapid as well, since even with no post-hybridization incubation for NCL reaction, significant antibody binding was observed (Figure 2D). Since NCL reactions could still occur during the binding step with antibody itself, an abbreviated 10 min time for trastuzumab binding was used (previously found to still allow measurable signal), which is taken as the effective shortest time point for NCL in this functional assay. No reactivity was seen using single haplomers alone (Figure 2B and Appendix A), again demonstrating the requirement for pairwise haplomer interaction for antibody recognition. 

As another control for the ELISA signal read-out, full-length mimotope peptides tagged with azides were reacted with the same DBCO-oligos #440 and #439 as used to prepare the trastuzumab-peptide haplomers **a** and **c,** respectively (Figure 1C and Appendix A). To contrast to the “half” status of haplomers, these complete peptide–oligonucleotide conjugates were termed “holomers” (α-holomer for the #439 conjugate, and β-holomer for the #440 conjugate). Both conjugates with the two separate oligonucleotides were capable of hybridization to the common biotinylated template used in ELISA assays, thus allowing reaction with trastuzumab, as was confirmed by experiment (Appendix A).

As well as using a solid-phase template, generation of the haplomer NCL product with a biotinylated DNA template could be performed in solution, followed by gel purification of the size-selected template–product complex. After elution of the complex, its subsequent capture in wells of a streptavidin plate allowed demonstration of trastuzumab reactivity by an ELISA assay (Appendix A). Templating was equally effective for generating an antibody-recognizable product when haplomers **[****c]** and **[****a]** (Figure 1) had their nucleic acid segments replaced with 2′-*O*-methyl RNA versions with the same sequences (Appendix A). When DNA haplomers **[****c]** and **[****a]** (Figure 1) were incubated in solution with a RNA version of the same template, the templated haplomer reaction product was resistant to a combination of RNase A and RNase H, as expected (Appendix A). 

Trastuzumab can thus recognize the reaction product between haplomers bearing modified fragments of a specific mimotope for this antibody, in a highly reproducible manner. Since neither of the fragments shows any measurable antibody reactivity until brought together by a complementary template, the primary aim of NATS towards templated reconstitution of a target for an immune mediator (antibody) is thus fulfilled in this instance. Considered together, the formation of a product recognizable by antibody, the generation of gel-resolvable product bands (Figure 2A and Appendix A), and the thiol/thioester structures of the peptide fragment precursors are all entirely consistent with the formation of the complete mimotope by well-established principles of NCL chemical reactivity (Figure 1A). 

### 2.5. Template Titration Effects In Vitro

From purely theoretical considerations, it was apparent that a binary haplomer system with template-mediated assembly is likely to be vulnerable to the effects of template excess, as depicted in Figure 3A. As a model for investigations of the effects of excess template on haplomer reactivity, we used an oligonucleotide template system with hybridizing shorter oligonucleotides modified with pyrene groups, where the latter modified oligonucleotides served as “pyrene haplomers”. When brought into molecular proximity, their pyrene moieties can (by stacking interactions) generate strong and readily measurable excimer fluorescence [39,40]. Thus, measurement of appropriate emission fluorescence in this system serves as a gauge for the extent of haplomer co-hybridization on a common template (Figure 3B, schematic depiction).

With limiting template, the fluorescent signal increased in a linear manner as a function of the molar ratio of pyrene haplomers (where the two haplomers were kept equimolar to each other), reaching a peak at template–haplomer equimolarity (Figure 3B,C). The fluorescent signal notably declined when template was in excess (Figure 3B). In contrast to the observed good linear curve for sub-equimolarity, the super-equimolarity relationship was clearly non-linear. It can be predicted that in template excess, the haplomer-elicited fluorescent signal should decline by a factor of E/N, where E = peak signal obtainable at equimolarity, and N = molar excess of template. A theoretical plot based on this premise conforms to a power curve as expected, but while the observed signals also conformed to a power relationship, they significantly deviate from the theoretical in showing a slower rate of decline (Figure 3C). Nevertheless, in conditions of high template excess, the signal diminution was marked.

Among factors that may influence the productivity of templated haplomer assembly, template secondary structure is important through its potential impedance of haplomer hybridization. Since the DNA template used for the pyrene haplomer tests (Figure 3) had a relatively low propensity for forming secondary structures (lowest ∆*G* = −3.31 kcal/ mol; Tm = 43.5 °C), we investigated an alternative system with a 30-mer RNA template (GAAAUAGAUGGUCCAGCUGGACAAGCAGAA) with a much higher secondary structure potential and predicted stability (lowest ∆*G* = −8.07 kcal/ mol; Tm = 71.7 °C). In this case, we used haplomers derivatized with linear alkyne and azide groups for enabling Cu(I) catalyzed click chemistry and visualized the resulting products on a denaturing gel (Appendix A). Since high levels of RNA template would interfere with gel analyses, we used akaline hydrolysis to remove RNAs after the completion of the click reactions. It was found that unlike the pyrene haplomer example, templated click product formation was stronger in conditions of template excess (even at a 10-fold template level), but at very high template molar ratios (100-fold) the reaction product levels were nonetheless markedly suppressed. A specificity control in the form of whole RNA from a melanoma cell line (MU89) showed no reaction product (Appendix A). 

### 2.6. “Locked-NATS” Design and Application for Overcoming Template Titration

The investigations of the reduction in haplomer product formation at elevated template levels reinforced the perception that a strategy should be defined for overcoming the template concentration effect, without which the full potential of the NATS process could not be realized. To this end, we developed the “locked-NATS” approach, based on structurally determined differential hybridizations. For these purposes, the two participating haplomers were redesigned such that one of them (the First or “bottle” haplomer) can assume a stem–loop structure, where the loop region comprises the complementary sequence to the nucleic acid target of interest. In the presence of specific target, hybridization with the loop occurs, which results in a strain force greater than the hydrogen bonding strength of the stem itself, thus favoring the opening of the structure, in an analogous manner to molecular beacons [41]. This in turn provides an accessible complementary sequence for the Second haplomer, as depicted schematically in Figure 4A for a target sequence derived from human papillomavirus (HPV). A consequence of this design is that, unlike the original haplomer templating process (Figure 1C), the hybridizing haplomeric nucleic acid moieties can be held constant in the Architecture 1 configuration (Figure 1B and Appendix A), while the target-complementary loop sequence may be varied in a completely modular fashion. Additionally, by design, the complementary hybridization sequence for the Second haplomer is only exposed when a First bottle haplomer molecule interacts with a molecule of specific template. Since excess template does not interfere with this, the interaction of the Second haplomer with the First bottle haplomer–template complex is predicted to be unimpeded by titration effects with excess template (further details supplied in Appendix A). 

Linear alkyne and azide groups were appended to the 5′- and 3′ ends of the first and second Locked-NATS haplomers, respectively (Figure 4A), in order to provide a functional read-out by means of the acceleration of copper-catalyzed click reactivity [42] via enforced molecular proximity. The copper-catalyzed click reactions were useful for these purposes in allowing precise control over the timing for the activation of click activity, unlike strained copper-free click alternatives that will occur spontaneously [38], especially when reactants are in molecular proximity.

Click product formation with the Locked-NATS haplomers was demonstrable under several different incubation conditions, but only in the presence of specific HPV DNA template (Figure 4B). Subsequently, we showed that a First haplomer with scrambled stem sequences did not support product formation (Appendix A), consistent with the need for hybridization between the “open” First haplomer stem sequence and the Second haplomer. In another test, a First haplomer was constructed that retained the hybridization segment complementary to the Second haplomer but lacked the complementary sequence that enabled stem-loop formation (Appendix A). Such an arrangement was expected to remove the “lock” on the hybridizing sequence for the Second haplomer in the absence of specific target, and indeed for this arrangement, product formation was independent of the desired target (Appendix A).

Since the Locked-NATS approach was predicted to potentially benefit from high template levels rather than be diminished, we used another Locked-NATS model with a known repetitive motif, in the form of the EBNA-1 transcript from the B95-8 Epstein–Barr viral genome [43]. For this purpose, the stem-complementary sequences (allowing the formation of the First haplomer “bottle”, Figure 4) were preserved, with the modular insertion of a varying loop sequence designed for the new target. This target sequence has 11 repeats within EBNA-1 coding sequence (15 if overlapping sequences are included, even more if limited base changes are included; Appendix A), and this was used for the design of an EBNA-1 model First haplomer (Appendix A). In the absence of template, no click product was seen (Figure 5; Lanes 1 and 2). Product was seen when the specific target EBNA1 sequence was present (Lanes 3–6), but not with a non-specific control oligonucleotide (GACTAGACGGCCAGGGAGACGAATACATATTCAAT, Lanes 7 and 8). (An apparent sequence-dependent mobility difference was observed for the EBNA1-specific template and non-specific control oligonucleotides, despite their being of identical lengths (35-mers)). Moreover, the click product yield was clearly present even with a 10-fold molar excess of specific target (Figure 5, Lanes 5 and 6).

In an additional assessment of template excess and Locked-NATS, we used an RNA HPV oligonucleotide compatible with the same Locked-NATS haplomers as used in Figure 4A. It was observed (Figure 6) that click products were promoted by the presence of the RNA template and were still present even when the template was in a 100-fold molar excess. These findings were compared with the click product formation from the same template, but with linear haplomers bearing the same click groups (Appendix A). In strong distinction to the Locked-NATS haplomers, click product formation with linear haplomers was essentially ablated after a 10-fold molar excess (Figure 6).

The utility of Locked-NATS is dependent on the generation of hybridization-enforced molecular proximity of haplomer 5′ and 3′ ends after “opening” of the First haplomer in the presence of target template. Since both the hexanyl- and azide click groups used for oligonucleotide modifications are themselves appended by C4/C5 alkyl linkers, molecular proximity and reaction acceleration is likely to be attained even if a tract of haplomer non-complementarity (Figure 4) is present near the reactive 5′ and 3′ ends. It was found (Appendix A) that Locked-NATS product formation was still effective when a 4-base end 3′-mismatch accompanied the same 12-bp hybridization segment as previously used (Figure 4). The 5′-ends of Second Locked-NATS haplomers could be extended with an additional arbitrary sequence tag without interference with product formation, potentially useful as a barcoding identifier of specific Locked-NATS combinations (Appendix A).

### 2.7. Locked-NATS and Pyrene Fluorescence

In the same manner as for standard linear haplomers equipped with pyrene tags (Figure 3), the utility of Locked-NATS could also be investigated with pyrene excimer fluorescence, thus providing a different system for testing its efficacy. For this purpose, 2′-*O*-methyl oligonucleotides were derivatized with pyrenes (Figure 7A and Appendix A), with the same First- and Second haplomer designs and HPV sequences as used for initial click-reaction testing (Figure 4 and Appendix A). When incubated with DNA template, the First pyrene haplomers can open and expose the site for subsequent hybridization with the Second pyrene haplomer (as for the depictions in Figure 4), allowing proximity-promoted excimer fluorescence. As the ratio of template to haplomers approached equimolarity, fluorescence increased in a linear manner (Figure 7B), increasing slightly to a 2-fold ratio, and then maintaining this level at least up to 20-fold template excess (Figure 7C). When the template dose responses of linear vs. Locked-NATS pyrene haplomers are compared as a % of equimolar template: haplomer fluorescence, the strong distinction between the two configurations is clearly demonstrated, along with the maintenance of signal strength independent of template levels with Locked-NATS (Figure 7D).

Pyrene excimer fluorescence can also be visualized in non-denaturing gel systems, which also affords a convenient means for assessing the relative efficiencies of variant nucleic acid templates with differing backbones. For this purpose, we compared the 2′-*O*-methyl RNA pyrene First haplomer used for template titrations (Lk-1068M; Figure 7) with a modified design with a reduced loop size directed at the same HPV target, and a stem region with complementarity for the Second haplomer Lk-1069M. (Despite their differing loops, both Lk-1068M and Lk-1120M can hybridize with a common target template, and subsequently use Lk-1069M as a common Second haplomer). These pyrene Locked-NATS First haplomers were separately incubated alone or with 5-fold excesses of 35-mer templates composed of DNA, RNA, and 2′-*O*-methyl RNA, also with in conjunction with Lk-1069M Second haplomer. When run on Tris-glycine non-denaturing gels, discrete bands showing Pyrene-mediated fluorescence could be observed directly without nucleic acid staining (Appendix A). Such fluorescence was only visible in sample lanes with both Locked-NATS haplomers and templates, and notably only with RNA or 2′-*O*-methyl RNA for Lk-1120M. Upon subsequent gel staining with SYBR-Gold, the template-mediated opening of the First haplomer “bottles” was directly evident, through the formation of new bands, superimposable with the direct-fluorescent bands noted before staining. In the presence of RNA or 2′-*O*-methyl RNA templates, both faster-migrating First haplomer stem–loop “bottle” bands were essentially absent through evident complexing (Appendix A). Although this was less apparent with the Lk-1068M First haplomer and DNA template, a new slower mobility band was observed, as well as the formation of direct pre-staining fluorescence. However, the Lk-1120M haplomer was evidently unable to open via the DNA template, showing neither significant new slower-mobility bands nor direct fluorescence.

### 2.8. Locked-NATS and NCL

Although pyrene haplomer excimer fluorescence is a convenient means for measuring template titration effects with respect to haplomer function, it does not involve covalent bond formation. Thus, we further extended Locked-NATS studies to exploit the same NCL product formation and trastuzumab antibody detection system as used for the initial conventional (linear) NATS (Figure 2, Figure 3 and Figure 4) and adapted this for a Locked-NATS strategy. Accordingly, we initially designed DBCO-modified oligonucleotides to function as First and Second Locked-NATS haplomers for conjugation to thioester and cysteinyl peptides with terminal azides, in an analogous manner as for linear NATS (Figure 2 and Figure 3).

For NCL application of Locked-NATS, we implemented another design variation for stem–loop formation, “shared stem”, previously applied with molecular beacons [44], where one arm of the stem is complementary to a part of the target sequence in common with the loop. This has the theoretical advantage of eliminating “sticky-end” intermolecular cross-hybridizations between opened forms of stem–loop structures [44]. The 5′-DBCO-modified Locked-NATS First haplomer for NCL application was designed with a 21-base loop and a “shared” 9-base stem, along with a corresponding 9-base Second haplomer bearing a 3′-DBCO modification (oligonucleotides Lk-HPV21Db and Lk-S11Db, respectively, Appendix A).

Following click conjugation reactions between the DBCO-modified Locked-NATS oligomers and azide-modified mimotope fragment thioester/cysteinyl NCL fragments (in a comparable manner to the procedures shown in Figure 1), preparations were desalted and used for testing the formation of the NCL product as a function of template concentration. Parallel tests were conducted with linear haplomers bearing the same active NCL fragments. Bands resulting from proximal NCL reactions were assigned as those both highly dependent on the presence of template and of appropriate size, for either linear or Locked-NATS reactions. (The apparent linear NATS NCL product size (~55 bases) is consistent with the observation that the migration of the DNA-peptide conjugates approximates to the summation of the base length of total DNA oligomers (16 + 17) + conjugated peptide amino acid residues (20 residues following NCL reaction, giving a total of 53). The observed size for the Locked-NATS NCL product (~65 bases) was smaller than that predicted by the simple DNA/peptide summation guide (71 total), but this is consistent with the faster mobility observed for other Architecture 1 NCL products (Figure 2, as described above). The apparent “negative staining” effect observed with high template levels is a SYBR-Gold staining artifact.

It was found (Figure 8A) that such an NCL-associated band for Locked-NATS remained observably constant even at 20-fold molar template excess, while a diminution of linear NATS product was apparent even at 5-fold molar template excess, consistent with the performances of the two NATS strategies in other systems (Figure 6 and Figure 7). We used the same NCL reactions to compare patterns of recognition by trastuzumab antibody in ELISA assays, where saturating amounts of biotinylated DNA templates were initially added to wells of a streptavidin plate (R&D Systems). This was followed by sequential Lk- or linear haplomer thioester/cysteinyl fragment conjugate hybridizations in a series of dilutions where equimolar amounts of both haplomer pairs were present at each dilution point. After incubations to allow NCL reactions, probing with trastuzumab antibody was performed in a standard manner (Figure 8B,C). Under such conditions, Locked-NATS reactions gave a good linear trastuzumab response from 5–20 pmol/well (Figure 8B), whereas linear NATS-derived signals were both considerably weaker and unable to fit to reasonable linearity over the same range, showing a sudden response drop below 15 pmol/well (Figure 8C). This effect can be interpreted as due to a template titration effect with the linear NATS, as the constant amount of bound template per well effectively is in excess when the applied haplomer amounts decrease below a certain threshold. Notably, this “sudden decline” effect was absent with Locked-NATS, evident from its linear response, and as predicted by its designed architecture (Figure 4).

During the generation of NCL products, it is important to avoid oxidation of the essential cysteinyl moiety. Addition of TCEP is effective in this regard, but its presence can potentially also result in chemical alterations of the desired traceless cysteine-bearing peptide product, resulting in desulphurization and conversion to dehydroalanine or even alanine itself under some circumstances [45]. Although complete conversion to alanine is unlikely with our NCL products (since no heating steps are used), it was not clear in any case whether such an alteration could potentially affect recognition by trastuzumab. Accordingly, we compared responses towards the original mimotope, the S11C derivative as produced by NCL (Figure 2 and Figure 3), and the corresponding S11A peptide. It was found (Appendix A), that although responses to the original mimotope and S13C were comparable (as in Appendix A), the recognition of peptide S11A was substantially reduced. Accordingly, to avoid potential sensitivity losses, sodium ascorbate (50 mM) was subsequently included in NCL reactions as a protective agent [46].

### 2.9. NATS-Mediated NCL Reactions on a Cell Surface for Antibody Recognition by Locked-NATS

To assess the assembly of a recognizable mimotope on a cell surface, we used human Jurkat cells treated with anti-Class I MHC antibody (W6/32) and streptavidin, followed by biotinylated template (Methods;Appendix A). With Locked-NATS haplomers, a reproducible signal was obtained by flow analysis following staining with trastuzumab antibody and fluorescent secondary antibody (Figure 9A). This template-dependent response was only obtained when both haplomers were added to treated cells, clearly demarcated from background signals with either haplomer component alone. The presence of surface template was also demonstrated by flow responses with a biFAM-labeled probe, along with its dependence on the primary W6/32 antibody and streptavidin (Figure 9B).

## 3. Discussion

In this study, we demonstrate a platform for in situ assembly of functional molecules for immune recognition, exemplified by a mimotope for the anti-HER2/neu antibody, trastuzumab. The functional formation of this NCL product is achieved by facilitating the combining of precursors (themselves unreactive with antibody) as directed by specific nucleic acid templates. Where the presence of target templates is specific to designated pathological cells, such a process has potential for future therapeutic purposes. In particular, it is an appealing strategy from its avoidance of “off-target” toxicity, if the template that directs the assembly process is absent or greatly reduced in normal cells. In principle, templates that can be utilized for NATS include virtually any accessible nucleic acid sequence, including viral, oncogene, translocations, point mutations, or non-coding sequences that are unique to specific tumor cells. Additionally, templates can be artificially tagged onto cells of interest by a variety of means, including the antibody–streptavidin system (Figure 9 and Appendix A) where the streptavidin functions simply as a useful adaptor for combining any biotinylated nucleic acid with an antibody of interest. Direct covalent modification of antibodies with nucleic acid templates is an alternative, or the tagging of templates with small-molecule ligands for specific surface receptors. The general NATS concept can be compared with pro-drug activation, where site-specific enzymatic catalysis generates active drug with a much more restricted and directed focus of activity than for general systemic application [47]. More generally, the NATS principle can be viewed as a conditional AND logic gate, where two inputs (a pair of co-designed haplomers) are required, but an output signal (functional read-out) is only attained in the presence of a specific template (Appendix A).

In order to assess the ability of the NATS strategy to direct the assembly of functional products, particularly those of immunological significance, we considered strategies for the reconstitution of short peptide sequences. While Staudinger traceless chemical ligation is bioorthogonal and potentially allows the formation of a wide range of native peptide sequences, in our experience the lability of the components (particularly the necessary modified phosphino-peptides as co-reactants) is a significant limiting factor. Consequently, we evaluated native chemical ligation (NCL) as an alternative means for achieving traceless peptide reconstitution. While this approach [36] has the limitation of requiring a cysteine residue at the site of peptide chemical ligation, its chemistry is robust under in vitro conditions.

Since antibody recognition of a cognate peptide epitope is easy to assay, we considered possible peptide targets of known antibodies, especially those of proven clinical significance. Many studies have shown that linear peptide sequences can be empirically obtained that mimic native epitope recognition [48,49]. Since known trastuzumab mimotopes [35,50] lack cysteine residues required for NCL, we screened one example (QLGPYELWELSH; [35]) for retention of significant trastuzumab binding after selected residue substitutions with cysteine and found that the variant S11C (QLGPYELWELCH) showed suitable activity. Appropriately modified peptide fragments of the S11C mimotope split at the cysteine residue did not inhibit trastuzumab activity, whereas mixtures of them did so following a pre-incubation period (Appendix A). Moreover, when the peptide fragments were conjugated with suitable oligonucleotides, reconstitution of recognition by trastuzumab was achieved in a template-dependent manner (Figure 2). Under the experimental conditions used, the NCL-mediated reconstitution of trastuzumab activity rapidly reached a plateau (Figure 2D), suggesting that the kinetics of the reaction are compatible with its potential therapeutic application. Template-assisted NCL reactions have been previously applied where the thiol exchange reaction transfers a template-tethered peptidyl thioester onto a second proximally tethered peptidyl group bearing a terminal cysteine. This transfer reaction results in the formation of the desired complete peptide tagged only to a single nucleic acid segment, which can be designed to promote reaction turnover at physiological temperatures [23,51]. In contrast, for the purposes of our study, the templated peptide mimotope remains tethered to flanking nucleic acid segments (Figure 2A and Appendix A) for display and recognition by the antibody trastuzumab. Immobilization of a templated peptide created by NCL may be advantageous in certain contexts (such as on the surface of a pathological cell of interest) for subsequent antibody targeting.

Initial tests with NCL-mediated NATS were designed as a proof of principle for the use of templated assembly to generate an immunologically relevant product. A potential disadvantage of traceless NCL reactions is their lack of exclusive bioorthogonality under in vivo conditions. Thus, the C-terminal thioester component of a reactive NCL pair may be subject to hydrolysis by widely expressed thioesterases, and *N*-terminal cysteines are potentially labile though oxidation or undesired thiol-mediated chemical reactions. Nevertheless, their potential utility in therapeutic NATS applications remains to be evaluated, since certain delivery strategies could minimize chemical damage during transit to target cells of interest, such as incorporation of NCL haplomers into protective liposomes or other delivery vehicles [52].

Despite the considerable promise of the NATS approach for future therapeutic purposes, a major potential limitation was evident at the outset, in the form of titration of separate haplomer constituents of a required functional pair when template levels are relatively high, particularly if there are limitations on the amount of haplomer that are deliverable to the pathological cells. This problem was both theoretically pertinent and confirmed by experiment (Figure 3). The observed deviation of pyrene haplomer behavior in conditions of template excess (E/N curves; Figure 3C) is consistent with single isolated (and unproductive) haplomer: template duplexes undergoing reassociation, such that the probability of mutual (productive) hybridization on the same template is increased over a steady-state where the single haplomer: template dissociation rate is low. Although the length and base composition of a targeted haplomer duplex may be adjusted for a desired melting temperature, in practice prediction of thermal stabilities and hybridization kinetics of both single haplomers and co-hybridized haplomer pairs before and after reactions is likely to be especially challenging for a therapeutic system applied intracellularly [53]. This is especially the case if proximal haplomer co-hybridization tends to be mutually stabilizing, as observed for experimental oligonucleotide hybridizations [54].

In the reverse condition of haplomer excess, product formation is limited by template availability (Figure 3C). Dissociation of reacted haplomer pairs (and product) from a template would in principle release the template for another hybridization/reaction round, and a perfectly recycling template in such circumstances would have a true and measurable catalytic role. In practice (particularly in a potential therapeutic setting) dissociation of reacted and linked haplomers from a template may not be rapid enough to promote multiple rounds of successive haplomer annealings. Another major potential influence on haplomer/template association is template secondary structure. The presence of significant secondary structures in an experimental template of interest changed the optimum haplomer/template ratios but did not remove the titration effect (Appendix A). Where equilibrium exists between template secondary structures that are inaccessible for haplomer hybridization and “open” linear arrangements that permit haplomer association, increased template levels concomitantly increase the available amount of accessible template. Nevertheless, at very high template concentrations, the titration effect still occurs, and product formation declines steeply (Appendix A).

These combined observations reinforce the importance of the template titration effect, while at the same time showing that prediction of the extent of product formation impedance simply by template concentration alone is not sufficient, even if that is practically achievable. Given these strictures on the general application of NATS, we sought a means to side-step the template titration problem, in the form of “Locked” (Lk)-NATS. Target sequence-determined differential hybridizations were shown to be effective in opening a short segment of a First “bottle” haplomer for hybridization by a Second haplomer, with concomitant formation of product from the two precursor components borne by the haplomer pair (Figure 4 and Figure 5). In such circumstances, the haplomer interaction conforms to Architecture 1 rather than Architecture 2 for conventional haplomer/template association (Figure 1C). Removal of the hybridizable tract in the First haplomer prevents reactivity (Appendix A), and removal of the “locked” stem structure stops the desired target-dependency of co-haplomer association (Appendix A).

In these studies, we made use of DNA, RNA, and modified (2′-*O*-methyl) RNA templates. Internal cellular targets for nucleic acid templating will invariably be RNAs (either mRNAs or non-coding), provided such templates are both accessible and constitute specific signatures for the target cells of interest. However, artificial nucleic acid templates placed on cell surfaces are not restricted in this regard, and indeed modifications towards nuclease resistance should be advantageous. It is known that nucleic acid duplexes with the same base sequences but different sugar or backbone modifications will vary in their melting temperatures (Tm values) [55]. Consequently, proof-of-principle for NATS and haplomers obtained in one nucleic acid format is predicted to be translatable into an alternative modified format of the same sequence, provided adjustments are made for known differences in Tm values, along with any propensities for secondary structure formation. This prediction is consistent with results from the present study, as effective generation of peptide NCL products recognized by trastuzumab was observed with DNA-based haplomers from DNA (Figure 2) or RNA templates (Appendix A) or 2′-*O*-methyl haplomers with DNA template (Appendix A).

Evidence for the resistance of Locked-NATS to titration effects by template excess was obtained in three model systems (click chemistry, pyrene fluorescence, and NCL; Figure 5, Figure 6, Figure 7, Figure 8 and Figure 9). In the system with a target sequence corresponding to a region of the HPV E6/E7 gene [56], product with Locked-NATS was observed even at 100-fold molar excess, but in contrast was extinguished after a 10-fold molar excess for a conventional (Architecture 2; Figure 1B) templating system (Figure 6). Removal of the constraints imposed through template concentration by Locked-NATS also can be seen as an enhancement of the assignment of NATS as a conditional AND logic gate (Appendix A), since the template-imposed conditionality is thereby extended from a relatively narrow range into much broader circumstances.

Additional studies of the Locked-NATS system are clearly warranted, including optimization of the loop region length and stem stability for achieving optimal target hybridization efficiency and specificity. Some initial work towards this end has been included in this study, where both stem and loop lengths were varied in different applications (see Appendix A for as a summary of Locked-NATS haplomer structural variables). All the DNA-based configurations were effective for the three types of read-outs used (Cu[I]-catalyzed click reactivity; pyrene excimer fluorescence, and NCL mimotope peptide formation). This includes the “shared-stem” configuration of the Locked-NATS First haplomer used for NCL (Figure 8 and Appendix A). Duplexes composed of 2′-*O*-methyl RNA are known to have markedly higher melting temperatures than corresponding DNA duplexes [57]. Accordingly, it follows that opening a 2′-*O*-methyl RNA stem–loop would require greater applied strain force mediated by loop hybridization than for a corresponding DNA stem loop. We tested pyrene-labeled First and Second haplomer 2′-*O*-methyl RNAs (Lk-1068M and Lk-1069M, respectively) with the same sequences as the initial HPV-directed DNA Locked-NATS system (Figure 4 and Appendix A). Pyrene excimer fluorescence was readily detectable with a DNA template, and at the same time showing the efficacy of the Locked-NATS system in template excess (Figure 7). In non-denaturing gels, First haplomer opening (mediated by hybridization with template) and subsequent interaction with Second haplomer is observable by (1) fluorescent band formation mediated by pyrene juxtaposition, (2) the appearance of slower-migrating bands corresponding to the formation of complexes with template, and (3) depletion of the fast-migrating stem–loop First haplomer bands. By these criteria, it was evident that opening of Lk-1068M is considerably more efficient with RNA or 2′-*O*-methyl RNA templates than DNA (Appendix A). This effect was magnified with the 2′-*O*-methyl RNA First haplomer Lk-1120M, where DNA template was unable to effect stem–loop opening or allow excimer fluorescence (Appendix A). Since Lk-1120M has a slightly shorter stem than Lk-1068M (Appendix A), it implicates the shorter loop length of the former (20 bases for Lk-1120M vs. 30 bases for Lk-1086M) in the failure of DNA template to open Lk-1120M. Evidently, the failure of DNA template hybridization to overcome the strength of the Lk-1120M stem duplex destabilized any prolonged interaction with the loop, as prominent new complexes were not observed on non-denaturing gels (Appendix A, Lane 8). On the other hand, the Lk-1120 stem loop was efficiently opened by corresponding RNA or 2′-*O*-methyl RNA templates (Appendix A, Lanes 9 and 10). These findings underscore the importance of the balance between stem stability and loop length for optimal Locked-NATS function. Since an important ongoing aim of this work is its application for intracellular RNA template targeting, future Locked-NATS development will include optimization of stem stability in the cytoplasmic environment, possibly using unnatural nucleic acids in the stem region, by analogy with molecular beacons [58].

By its design, productive haplomer duplexing in the Locked-NATS system is based on a bimolecular Architecture 1 interaction (Figure 1B and Appendix A) through a single short duplex that may be held constant while the loop region for target hybridization is varied in a modular fashion as required (Figure 4A). The Locked-NATS process thus has an inherent functional separation between hybridization-mediated target sequence recognition and the hybridization between the First and Second haplomers that follows “unlocking”, in turn allowing product formation. Since the fixed sequences in the “unlocked” configuration that enforce haplomer proximity are independent of target recognition and relatively short, the opportunity arises for systematic investigations of sequence modifications designed to enhance turnover, possibly by similar strategies as previously described [23,51].

To become therapeutically effective, a NATS/haplomer system requires identification or provision of a useful template in the context of a cellular target. Many pathological cells provide a wealth of potential target-specific intracellular RNA molecules, but exploiting such templates is simultaneously associated with delivery challenges. Where surface targets are chosen for haplomer templating, a template oligomer may be placed on the surface by an antibody-based system (Figure 9 and Appendix A), or potentially by conjugation with an “anchor” ligand for a known surface receptor that demarcates cell targets of interest. We have shown here that antibody immune recognition is achievable through splitting a target peptide into inert fragments and assembling them on a nucleic acid template via specific chemical reactions. It is anticipated that many therapeutic applications for this NATS process will emerge in the future, particularly for converting otherwise negative or weakly antigenic cells into immunologically recognizable targets. Furthermore, beyond the creation of immune targets, in principle a wide variety of biologically active molecules can be split into inert moieties and then functionally assembled via NATS, including toxins, chemo-attractants or other therapeutically useful entities.

In principle, NATS allows assembly of the same product molecule on different templates, or different products on the same template, both of which have practical implications. Thus, where expression of a primary tumor template nucleic acid is lost through selective pressure, identification of additional templates in the resulting tumor variants may overcome such loss-mediated evasion, thereby allowing continued assembly of the original therapeutic product. Moreover, it may also be feasible to simultaneously target multiple sequences in the same tumor cells with several different haplomer pairs, to forestall tumor escape by specific template loss. The NATS strategy accordingly offers the possibility of novel pathways for effective tumor therapies, as well as for other diseases where the elimination of pathological cells would prove beneficial.

## 4. Materials and Methods

### 4.1. Oligonucleotides, Peptides and Cell Lines

Biotin-, azide- and alkyne-modified oligonucleotides were purchased from International DNA Technologies (IDT, Coralville, IA, USA); pyrene oligonucleotides from TriLink Biotechnologies (San Diego, CA, USA). Thioester and azide-modified peptides (>90% purity by HPLC) were supplied by Anaspec (Fremont, CA USA). Further details of peptide and oligonucleotide syntheses are provided in Appendix A. The melanoma cell line MU89 was obtained from human melanoma tissue as described [59]; Jurkat cells were obtained from ATCC.

### 4.2. Cu(I) Catalyzed Click Reactions for Linear Alkynes and Azides

Click oligos (typically 50 pmol each for 5′-azide and 3′-linear alkyne-labeled strands) and various templates (50 pmol) were initially annealed in 25 µL 10 mM Tris pH 7.5/10 mM MgCl_2_/50 mM NaCl/1 mM dithioerythritol by heating 2′ 80 °C and cooling to room temperature. From each annealing, 10 µL (20 pmol) was then subjected to Cu(I) click catalysis or kept in equivalent buffer without catalysts. A premix of the following components was prepared with additions in the following order: 20 µL 70 mM Tris (3-hydroxypropyltriazolyl methyl) amine (THPTA, Sigma-Aldrich, St. Louis, MO, USA) in 0.155 M NaCl, 4 µL 500 mM sodium ascorbate in 0.155 M NaCl, 2 µL 100 mM CuSO_4_ in 0.155 M NaCl. From this premix, 2.6 µL was then added to each of the tubes for the click reaction, such that the final volume was 50 µL in 1× phosphate-buffered saline. Tubes were incubated for 30 min/0 °C (on ice), and then 2 h/25 °C. At the end of the incubation period, tube contents were desalted though Bio-Rad P6 columns as above and precipitated with 20 µg glycogen (Sigma-Aldrich, St. Louis, MO, USA). After centrifugation, pellets were washed with 70% ethanol, dried, and redissolved in 5 µL TE. Samples of each (1 µL) were run on 15% urea denaturing gels and stained with SYBR-gold.

### 4.3. Pyrene Haplomer Testing and Templates

For linear NATS, oligonucleotides with either 5′ or 3′ pyrene modifications (TriLink, San Diego, CA, USA) were incubated together with or without varying amounts of template or control oligonucleotides in phosphate buffered saline (pH 7.5, 150 mM NaCl) at desired molar ratios. After 1 h hybridization time, preparations were transferred to black-sided flat-bottomed 96-well plates (Corning; replicate wells per each sample), and fluorescence was measured in a Tecan spectrophotometer with an excitation wavelength of 485 nm and emission wavelength 535 nm. Predicted template secondary structures and analyses were derived from UNAFold (unafold.org) via licensed copy to IDT Coralville, IA, USA. For Locked-NATS tests, the First pyrene haplomers were initially heated in TN buffer (50 mM Tris pH 7.5/100 mM NaCl) 3 min/80 °C, and immediately chilled on ice for ≥15 min to promote intramolecular stem–loop formation. Constant amounts of First pyrene haplomers were incubated with varying amounts of template at desired molar ratios for 1.5 h/37 °C, before addition of the Second pyrene haplomer for an additional 30 min/37 °C. Fluorescence was then measured in the same manner as for linear NATS haplomers.

### 4.4. Base Treatment of RNA Templates and Gel Testing of Templated Click Products

Following click reactions or control treatments, all participating oligonucleotides in RNA templating reactions were desalted on Microspin P6 columns (BioRad) and treated with 0.2 M NaOH for 20 min/70 °C. Such preparations were then neutralized with acetic acid and 1 M Tris pH 7.4 and precipitated with 20 µg glycogen/0.3 M sodium acetate/3 volumes of ethanol. Spun pellets were washed with 70% ethanol, dried, and reconstituted in 10 mM Tris pH 7.4/0.1 mM EDTA, prior to formamide denaturation at ≥95 °C and loading on 8 M urea 19:1 bis-acryamide gels. Gels were stained with SYBR-Gold (Thermofisher, Waltham, MA, USA) and visualized under UV transillumination. In studies requiring excision of bands from gels, the procedure was followed as described in [60].

### 4.5. ELISA Assays for Trastuzumab

For all assays, streptavidin-coated plates (96-well, R&D Systems) were initially pre-washed twice (200 µL per well, PBS-T), and appropriate wells then incubated with biotinylated target peptides in dilution series. After 2 h/RT, wells were washed ×5 in PBS-T, and trastuzumab (BioVision) was added at the desired dilution (usually 2 ng/µL). Plates were incubated 1 h/RT, washed ×2 as before, and then incubated with the secondary antibody, an HRP conjugate of anti-human kappa light chain antibody (Southern Biotech). After an additional 1 h/RT incubation followed by x6 washes, enzyme activity was monitored by standard TMB reagents (Thermofisher, Waltham, MA, USA) for 30 min/RT and stopped with the addition of 100 µL 0.1 M H_2_SO_4_. Signals were quantitated at 450 nm and adjusted for blank buffer-only controls.

Where ELISA assays were the read-out for in situ templated NCL reactions, a biotinylated template oligonucleotide was added to streptavidin plate wells in excess (100 pmol/well) for 2 h/RT, before proceeding with the addition of single templatable NCL-reactive haplomers.

### 4.6. Template Positioning by MHC Class I Antibody and Streptavidin, and NCL on Cell Surfaces

As a convenient means for placement of nucleic acid template strand on a cell surface, we used a system with biotinylated antibody against a defined cell-surface marker, followed by treatment with tetravalent streptavidin and subsequently the biotinylated template of interest. The specific antibody chosen was the W6/32 monoclonal directed against human MHC Class I. A biotinylated derivative of W6/32 (Thermofisher, Waltham, MA, USA) was used to treat harvested and washed Jurkat cells in PBS at 10^6^ cells/mL (10 µg Ab/10^6^ cells) for 30 min 0 °C, washed, and then treated with biotinyated template (typically biotin—AGCTGTGTCCTGAAGAAAAGCAAAGACATCTGGACAA) in PBS (80 pmol/10^5^ cells) for 30 min 0 °C. Following washing, Lk- or linear haplomers derivatized by click reactions with thioester/cysteinyl mimotope fragments were added to 5 × 10^5^ resuspended cells in 0.5 mL (0.5 pmol/µL for each haplomer) for 1 h/22 °C. Following haplomer treatments, cells were stained with trastuzumab antibody (BioVision), followed by a secondary FITC-labeled goat anti-human kappa chain antibody (Southern Biotech) that does not bind to the primary murine W6/32 monoclonal. As a fluorescent control for the presence of template, a complementary oligonucleotide bearing 6-FAM groups at both 5′ and 3′ ends (6Fam- TCCAGATGTCTTTGCTTTTCTTCAGGACACA -6Fam) was used to hybridize with template-bearing cells. Cells were finally washed and subjected to flow analysis as described [59]. The template-positioning process and subsequent haplomer hybridizations are depicted schematically in Appendix A.

## 5. Conclusions

The nucleic acid templated synthesis (NATS) systems described in this work build upon previous studies integrating nucleic acid-templated chemistry with resulting product formation that is effective at low concentrations, such as formation of epitopes for highly sensitive antibodies. NATS compounds efficiently produce an active trastuzumab mimotope selectively in the presence of a specific nucleic acid template. Templated production of antibody epitopes introduces a potent and versatile new class of product molecules to the field of nucleic acid-templated chemistry.

A constraint on NATS utility exists in the form of titration effects, which dramatically diminish product yield when template is present in a high molar excess relative to NATS reactants. These effects can be particularly limiting in the context of live cells, where mRNA templates may be present at a significant molar excess relative to the amount of NATS reactants that can be effectively delivered to the cytosol.

“Locked-NATS” was developed to circumvent titration-related reduction in product yield. The Locked-NATS design motif offers multiple advantages:

Mitigation of titration effects;Maintenance of product yield over multiple orders of magnitude of relative template concentration;Simplified compound design.

Utilizing this approach, sequence-templated generation of trastuzumab epitopes could be detected even on the surface of live cells. Thus, Locked-NATS represents an advance towards utilizing sequence-triggered templated chemistry in complex biological contexts. Future work will focus on applying Locked-NATS compounds to probing clinically relevant nucleic acid sequences within living cells.

## 6. Patents

US Patent 10,603,385. (Issued 31 March 2020.) Methods and compositions for templated assembly of nucleic acid specific heterocompounds. Dunn, I. and Lawler, M.

US Patent 11,1868,39. (Issued 30 November 2021.) Methods for preventing titration of bimolecular templated assembly reactions by structurally-determined differential hybridizations. Dunn, I. and Lawler, M.

## Figures and Tables

**Figure 1 molecules-27-06831-f001:**
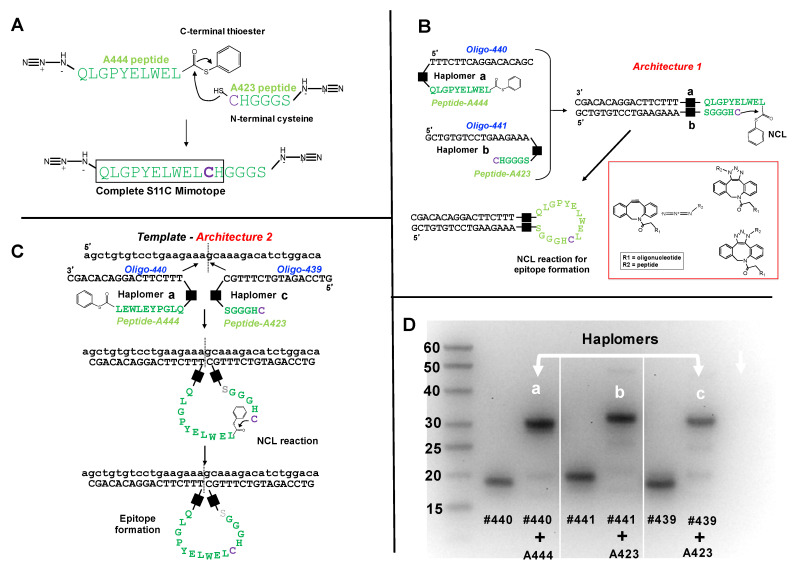
Trastuzumab haplomer structures and analyses. (**A**) Structures of the trastuzumab S11C mimotope split into two NCL-activated peptide segments N-terminal to the cysteine residue, where the larger A444 peptide bears a C-terminal phenyl thioester, and the C-terminal peptide has N-terminal cysteine. Joining the two peptides by NCL recapitulates the original S11C peptide. (**B**) Schematic depiction of Architecture 1 with mutual hybridization of haplomers **[a]** and **[b]** for assembly of trastuzumab mimotope S11C, and their mutual hybridization in Architecture 1. Black squares in Figure 1B,C denote the triazole linkage resulting from the azide-DBCO reactions, shown structurally in the red subpanel box. (**C**) Schematic depiction of Architecture 2, with haplomers **[****a]** and **[****c]**, and their hybridization on adjacent template sites for assembly of trastuzumab mimotope S11C. (**D**) Formation of haplomers **[a]**–**[c]** as shown with a 15% denaturing urea gel, with molecular weight markers (bases) indicated. Haplomer [a] was formed when N-terminally azide-modified peptide A444 (Figure 1A) was reacted in a 20-fold molar excess in sodium phosphate/100 mM NaCl buffer pH 7.0 with 5′- dibenzocyclooctyne (DBCO)-modified oligonucleotide #440 (DBCO-TTTCTTCAGGACACAGC) for 16 h. Haplomers **[b]** and **[c]** were respectively formed when C-terminally modified peptide A423 (Figure 1A) was incubated under similar conditions with 3′- DBCO-modified oligonucleotide #441 (GCTGTGTCCTGAAGAAA-DBCO) or 3′- DBCO-modified oligonucleotide #439 (GTCCAGATGTCTTTGC-DBCO). Gel lanes show relative migrations of DBCO-oligonucleotides alone (#440, #441, and #439) or after DBCO-oligonucleotide click reactions with azide-modified peptides (each DBCO-oligonucleotide + A444 or + A423, as shown). Further peptide details are provided in Appendix A, and additional details for oligonucleotides #440, #439, and #441 are provided in Appendix A.

**Figure 2 molecules-27-06831-f002:**
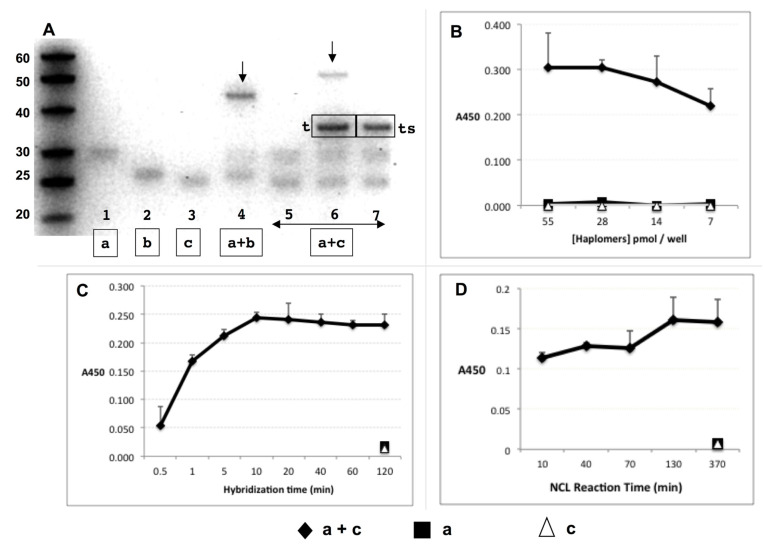
Trastuzumab haplomer reactivities. (**A**) Formation of NCL reaction products via haplomer templating, with 30 pmol haplomers 20 µL/16 h incubations in phosphate buffer pH 7.5 with 1 mM TCEP, where 1.5 pmol of each haplomer (structures as in Figure 1) is present per 15% denaturing gel lane. Haplomer codes correspond to structures in Figure 1 (haplomer [a] = oligonucleotide #440 click conjugated to peptide A444; haplomer [b] = oligonucleotide #441 click conjugated to peptide A423; haplomer [c] = oligonucleotide #439 click conjugated to peptide A423). Lanes 1–3: haplomer **[****a]** and reduced haplomers **[****b]** and **[****c]** alone, respectively; Lane 4: mutually complementary **[****a]** + **[****b]**, with product band shown by arrow, Lane 5-7: **[****a]** + **[****c]** haplomers alone; in the presence of equimolar specific mutual template (marked [t] in gel), or with corresponding scrambled template (marked [ts] in the gel), respectively. Product band only observed with correct template is also shown by arrow. (**B**) Recognition of NCL product of templated haplomers **[****a]** + **[****c]** by trastuzumab in ELISA assay. Excess biotinylated template oligonucleotide (1 pmol/µL, sequence as for Figure 1) was bound to streptavidin-coated ELISA plate wells, washed, and then incubated with haplomers **[****a]** and **[****c]** (30 pmol) either separately or together for 16 h. Plates were then washed and incubated with trastuzumab, followed by treatments with HRP-conjugated secondary anti-kappa chain antibody, with final enzyme substrate for signal development. (**C**) Hybridization kinetics of haplomer-template binding in ELISA assay. With streptavidin ELISA plates coated with biotinylated template oligonucleotide and washed, mixes of haplomers **a** and **c** (in PBS-1 mM TCEP; 25 pmol each; 100 µL total) were added to replicate plate wells for the indicated time periods (up to 2 h) after which the supernatants were removed, and wells washed twice more with PBS-TCEP. Control single-haplomers alone were incubated for the longest (2 h) time-point. Following all washes, wells were incubated for 16 h to allow NCL reactions to proceed to completion, and trastuzumab signals assayed as above. (**D**) Kinetics of NCL reaction in ELISA assay. Conditions of haplomer concentrations and template binding in streptavidin plates were as for Figure 2B, with a 10 min in-well hybridization time for each NCL reaction time point, followed by 3 washes and incubation in 100 µL PBS-TCEP. Haplomers for each time point were added on a staggered basis with decreasing incubation times for NCL reactions, until the first time point when washing and antibody screening followed immediately after the initial hybridization. Here, the screening was performed by replacement of well contents with trastuzumab dilution in PBS (2 ng/µL), and incubation for 10 min only before subsequent washing and signal development. Since in this functional assay NCL reactions can potentially continue during the abbreviated antibody binding period, the shortest time point for NCL reactivity is designated at 10 min.

**Figure 3 molecules-27-06831-f003:**
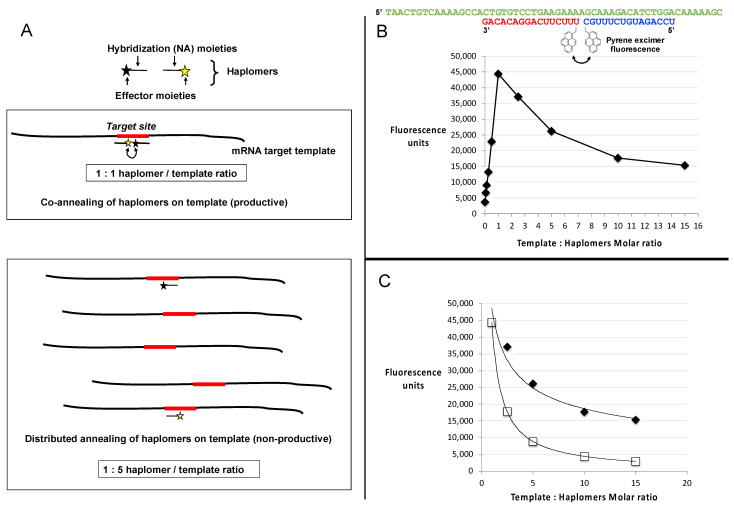
Template titration effects in vitro. (**A**) Schematic depiction of a binary haplomer pair and a template conforming to Architecture 2 (as shown in Figure 1C and Appendix A), in equimolar conditions allowing mutual hybridization and reaction to occur (1:1 haplomers/template ratio panel), and in conditions of template excess (1:5 haplomers/template ratio) where haplomer hybridization is stochastically favored to occur on separate templates as a function of relative template excess. When haplomer co-hybridization on a mutual template is thus disfavored, product formation similarly falls. (**B**) Generation of excimer fluorescence by pyrene-labeled 2′-*O*-methyl oligonucleotides (pyrene haplomers) mutually complementary on a common template (sequences shown at top), over a range of template concentrations. The part of the curve corresponding to haplomer excess (template: haplomer ratio < 1), to the point of mutual template/haplomer equimolarity shows a linear relationship (r^2^ value = 0.9986). (**C**) Power curves for pyrene haplomer fluorescence in conditions of template excess (data as for Panel B), for both measured (black diamonds, curve conforms to y = 48530.x^−0.416^; r^2^ value = 0.9681) and theoretical datapoints (open squares). The theoretical curve is calculated as a function of the optimal equimolar maximum E with the template molar excess N, as E/N (curve thus conforms to y = E.x^−1^). Standard deviations for data points were <5% of average values from replicate determinations.

**Figure 4 molecules-27-06831-f004:**
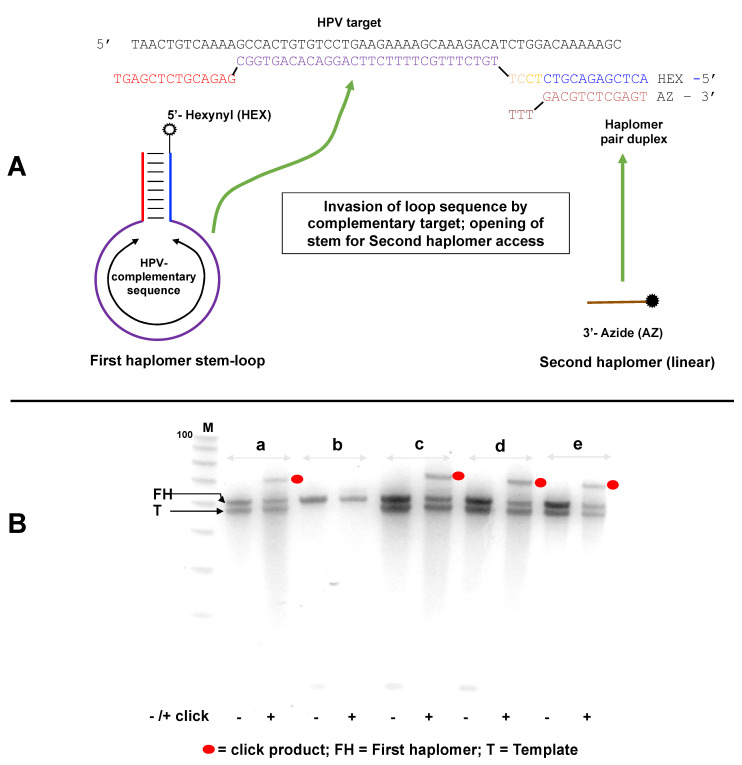
Locked-NATS strategy for overcoming template titration. (**A**) Design and structure of a Locked-NATS system. The First (or “bottle”) haplomer (Lk-HPV1) consists of a constant stem self-hybridizing duplex defining a variable loop sequence, which comprises the complement to any designated target (shown here as a DNA equivalent to a sequence from human papillomavirus (HPV)). The 5′ end of the first haplomer is modified to bear a reactive linear hexanyl group. In this example, a shorter oligonucleotide, the Second (linear) haplomer (Lk-HPV2), has a 3′ azide modification. In the presence of specific target template, hybridization occurs which destabilizes (“unlocks”) the stem duplex, exposing a single-stranded region that is complementary to the Second haplomer. Hybridization of the second haplomer with the “open” First haplomer results in the hexanyl/azide groups falling into spatial proximity, promoting their mutual click reactions in conjunction with Cu(I) catalysis. (**B**) Locked-NATS demonstrations with template/haplomer system of A under varying reaction conditions (a–e), where each is finally treated with and without catalysts for promoting Cu (I) click reactions. Before use, the First haplomer bottle was subjected to a “pre-self-annealing”, to ensure maximal intramolecular formation of the bottle structure. For this purpose, 1250 pmol of the first HPV haplomer bottle in 25 μL in 10 mM Tris-HCl pH 7.5/10 mM MgCl_2_/50 mM NaCl/1 mM dithioerythritol was subjected to 5 min/80 °C treatment, followed by rapid chilling at 0 °C for at least 10 min. From this, 1 μL (50 pmol) was used for the experimental tests. When the second haplomer was added, it was at an equimolar concentration with the first haplomer bottle (2 μM). Conditions: (a) First haplomer + target template incubated 2 h/37 °C; then second haplomer added for an additional 1 h/37 °C, (b) as for (a) without template, (c) First haplomer + target template incubated 2 h/25 °C; then second haplomer added for an additional 1 h/37 °C, (d) First haplomer + target template incubated 5′/80 °C, then subjected to a slow cool to room temperature, and then second haplomer added for an additional 1 h/37 °C, (e) all components (first haplomer, target template, second haplomer) incubated 1 h/37 °C (no pre-incubation of first haplomer and target). Following each incubation, preparations were treated with or without reagents for click reactions, processed by desalting, and samples analyzed on a 10% 8 M urea denaturing gel and stained with SYBR-Gold. Black dots = click products; FH = first (initial stem loop) haplomer; T = template. The small Second haplomers (staining weakly with SYBR-Gold) are visible near the bottom of the gel (indicated by arrow). M = 100/20 molecular weight markers (IDT; 10-base decrements from 100 bases). Further oligonucleotide details for Lk-HPV1 and Lk-HPV2 are provided in Appendix A.

**Figure 5 molecules-27-06831-f005:**
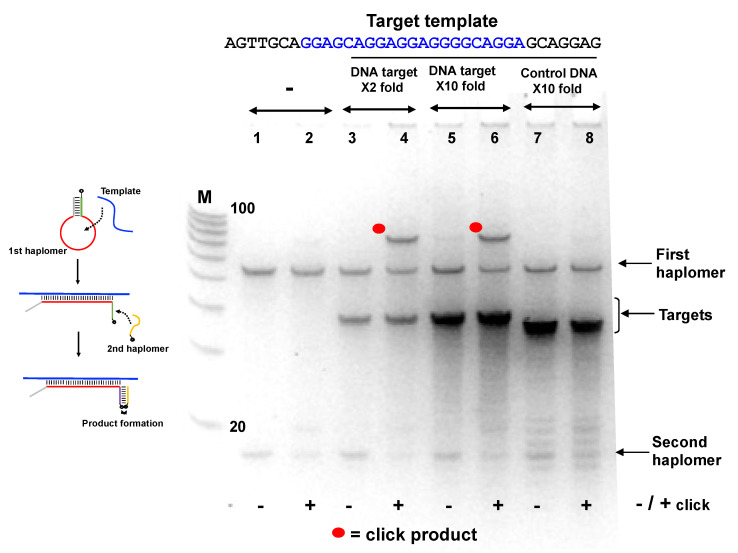
Locked-NATS strategy in conditions of template excess, using the EBNA1 First haplomer bottle (Lk-EBV1; 5′-hexanyl-TTCGACTCGAGACGTCTCCTTCCTGCCCCTCCTCCTGCTCCGAGACGTCTCGAGT) and second haplomer (Lk2; GACGTCTCGAGTTCTT-3′-azide) with a DNA oligonucleotide corresponding to the EBNA1 repeat sequence (“Target template”; sequence as shown with the central region complementary to the First haplomer loop in blue text). The EBNA1 first haplomer bottle was initially pre-self-annealed in the same manner as for the First haplomer bottle in Figure 4. Samples of the self-annealed first haplomer bottle were then incubated in 10 mM Tris-HCl pH 7.5/10 mM MgCl_2_/50 mM NaCl/1 mM dithioerythritol at a final concentration of 2 µM, along with the DNA template at both 2-fold and 10-fold excess concentrations. In addition, a control non-specific DNA 35-mer oligonucleotide (GACTAGACGGCCAGGGAGACGAATACATATTCAAT) was used, also at a 10-fold excess concentration. After a 2 h/37 °C incubation, the second haplomer was added to a final concentration of 2 µM, and the incubation continued for a final 1 h/37 °C. Following this, 10 µL samples from each test were subjected to +/− click reactions in the same manner as for Figure 4. Samples (1 µL) were analyzed on 10% 8 M urea denaturing 10% (19:1) acrylamide gels after denaturation in 98% formamide at 98 °C/3 min and immediate transfer to ice. M = 100/20 molecular weight markers (IDT; 10-base decrements from 100 bases). Further oligonucleotide details for Lk-EBV1 and Lk2 haplomers are provided in Appendix A.

**Figure 6 molecules-27-06831-f006:**
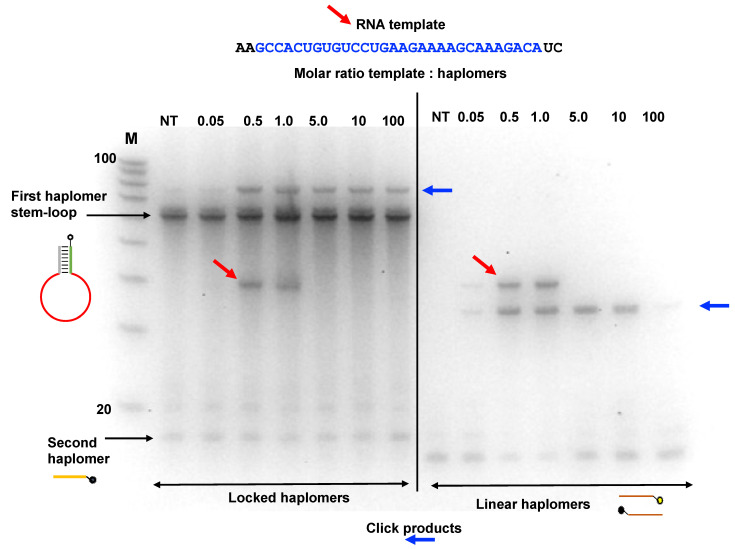
Locked-NATS strategy in conditions of template excess with an RNA template. The Locked-NATS click oligonucleotides as used in Figure 4 (First haplomer: 5′-hexanyl-ACTCGAGACGTCTCCTTGTCTTTGCTTTTCTTCAGGACA CAGTGGCGAGACGTCTCGAGT; Second haplomer: TTTGACGTCTCGAGT-3′azide) were tested with an RNA target template molecule (as shown by diagonal red arrows) corresponding to a truncated version of the HPV template DNA strand as also used in Figure 4. The same conditions as for Figure 4 were used with the First haplomer bottle in terms of quantities (50 pmol; 2 pmol/µL during template and second haplomer hybridizations; 0.4 pmol/µL during click reactions) and incubation times (2 h first haplomer bottle/RNA template annealings; 1 h for subsequent incubations with Second haplomer). Click reaction products are indicated with horizontal blue arrows. Locked-NATS reactions were compared with linear click haplomers directed to the same template (Lin-HPV1: 5′-hexanyl-TCAGGACACAGTGGC; Lin-HPV2: TGTCTTTGCTTTTCT-3′azide; detailed in Appendix A). The RNA template was used in a range of molar ratios ranging from 0.05:1 (template:haplomers) to 100:1. Where the amount of RNA was equal to or greater than 5:1, it was necessary after the +/− click reactions (performed in an identical manner as for Appendix A) to remove the template with alkaline hydrolysis (Methods) prior to gel analysis, to avoid interference with band patterns. Samples (1.0 µL) were run on denaturing 15% 8 M urea gels. NT = no template; M = 100/20 DNA size markers (IDT; 10-base decrements from 100 bases).

**Figure 7 molecules-27-06831-f007:**
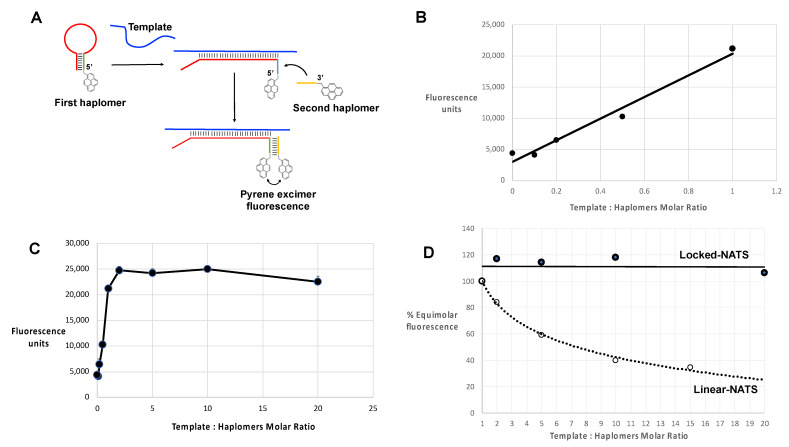
Locked-NATS strategy in conditions of template excess, using pyrene-labeled 2′-*O*-methyl RNA haplomers in an analogous manner to the linear examples of Figure 3, and as described in Methods, with a 30-mer template GCCACTGTGTCCTGAAGAAAAGCAAAGACA. The First stem–loop pyrene haplomer was Lk-1068M (2′-*O*-methyl RNA ACUCGAGACGUCUCCUUGUCUUUGCUUUUCUUCAGGACACAGUGGCGAGACGUCUCGAGU), and Second pyrene haplomer was Lk-1069M (2′-*O*-methyl RNA UUUGACGUCUCGAG) (further details of these are in Appendix A). (**A**) Schematic structures and function of pyrene Locked-NATS haplomers; (**B**) Pyrene excimer fluorescence from 0 to equimolar template showing linear response (r^2^ = 0.9747); (**C**) Pyrene excimer fluorescence with template up to 20-fold excess; (**D**) Comparison between linear-NATS (Figure 3) and Locked-NATS pyrene excimer fluorescence as a function of template concentration, expressed as a % of the fluorescence attained at template:haplomer equimolarity.

**Figure 8 molecules-27-06831-f008:**
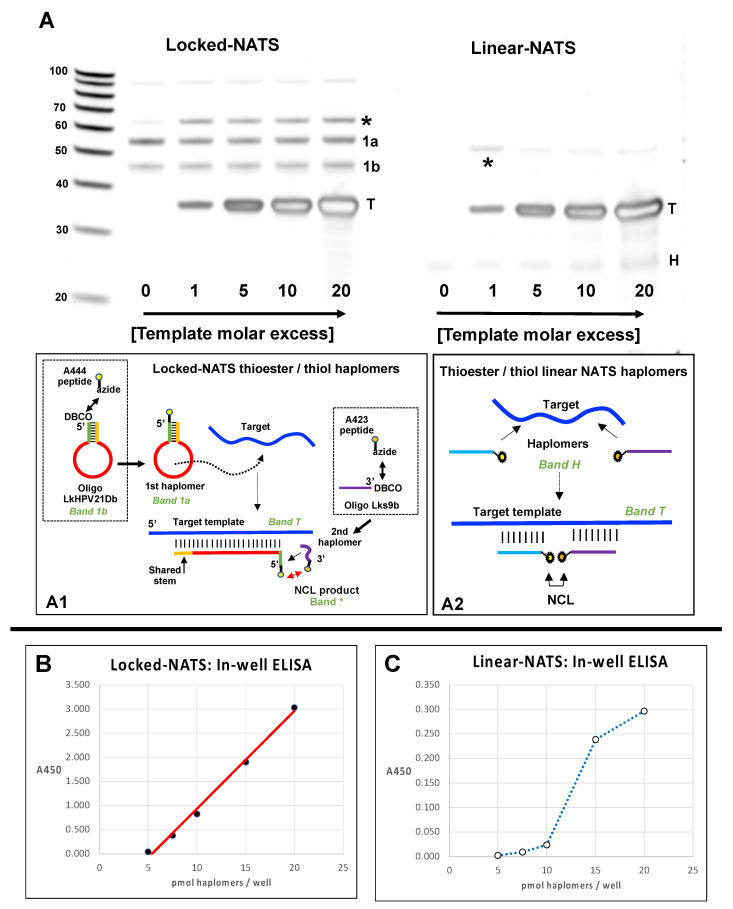
Locked (Lk) NATS strategy vs. Linear NATS with NCL assembly of a trastuzumab mimotope. (**A**) Denaturing 15% urea gels with Lk- and linear NCL reactions without template (0) and from 1–20-fold template excess. The Locked-NATS dibenzocyclooctyne (DBCO) oligonucleotide LkHPV21Db (DBCO-GCCACTGTGCTTTGTCTTTGCTT TTCTTCAGGACACAGTGGC) and the linear DBCO oligonucleotide 440 (Figure 1; DBCO-TTTCTTCAGGACACAGC) were conjugated with the trastuzumab azide- thioester mimotope fragment peptide A444 (Figure 1; Azide-SGGGQLGPYELWEL-[phenyl thioester]; the Locked-NATS DBCO oligonucleotide Lks9b (CACAGTGGC-DBCO) and the linear DBCO oligonucleotide 439 (Figure 1; GTCCAGATGTCTTTGC-DBCO) were reacted with the cysteinyl-azide mimotope fragment peptide A423 (Figure 1; CHGGGSK-azide). Conjugation reactions for the Locked-NATS oligonucleotides are depicted in dotted squares of subpanel A1. After conjugation reactions (as for Figure 2; 10-fold molar peptide excess), thiol reactions were reduced with 10 mM TCEP (Tris-(2-carboxyethyl) phosphine), and buffer-exchanged into PBS/1 mM TCEP/50 mM sodium ascorbate (PTA buffer). Paired Lk- and linear NATS conjugates were incubated for NCL reactions (3.75 µM; 1 h/22 °C) in PTA buffer with varying molar ratios of HPV DNA template Temp35.2 (GCTGTGTCCTGAAGAAAAGCAAAGACATCTGGACA) complementary to the loop sequence for the Locked-NATS First haplomer, or for both linear NATS haplomers for hybridization in a contiguous arrangement as indicated. Samples (3.75 pmol) were then tested on 15% acrylamide 8 M urea denaturing gels and stained with SYBR-Gold. Depicted schematically below the gel images are target template interactions of a Locked-NATS stem–loop with a shared-stem configuration (as for First haplomers formed with oligonucleotide LkHPV21Db), and conventional linear NATS. For both sets, asterisks indicate NCL-associated bands, T = template (depicted in subpanels **A1** and **A2**). Locked-NATS set; 1a = DBCO First haplomer click-modified with azide partial mimotope thioester; 1b = residual unreacted First haplomer (depicted in subpanel **A1**). The free 9-mer small Second haplomer for Locked-NATS stains poorly with SYBR-Gold and is not visible on the Locked-NATS gel set. Linear NATS set: H = co-migrating unreacted modified haplomer pair (depicted in subpanel **A2**). (**B**) Trastuzumab ELISA for the same set of Locked-NATS reactions in a streptavidin plate loaded with saturating amounts of 5′-biotinylated HPV template. Indicated molar amounts of the partial mimotope thioester-modified First haplomer were added to wells for hybridization (PTA buffer]) for 1 h, the supernatant removed, and after washing with PBS, the same molar amounts of the Second (partial cysteinyl mimotope) haplomer were added and hybridized under the same conditions as for the First haplomer. Supernatants were then replaced with fresh PTA buffer alone and incubated for 1 h/37 °C for NCL reactions to occur. Subsequently, wells were probed with trastuzumab, and developed with HRP-modified secondary anti-kappa antibody (Methods). Between 5–20 pmol of haplomers/well, a good linear trastuzumab signal was seen (r^2^ = 0.994). (**C**) Trastuzumab ELISA for the same set of linear NATS reactions, performed under the same conditions as for Locked-NATS, with the thioester-modified linear haplomer hybridized first, followed by the cysteinyl linear haplomer. Further peptide, oligonucleotide, and Locked-NATS details are provided in Appendix A.

**Figure 9 molecules-27-06831-f009:**
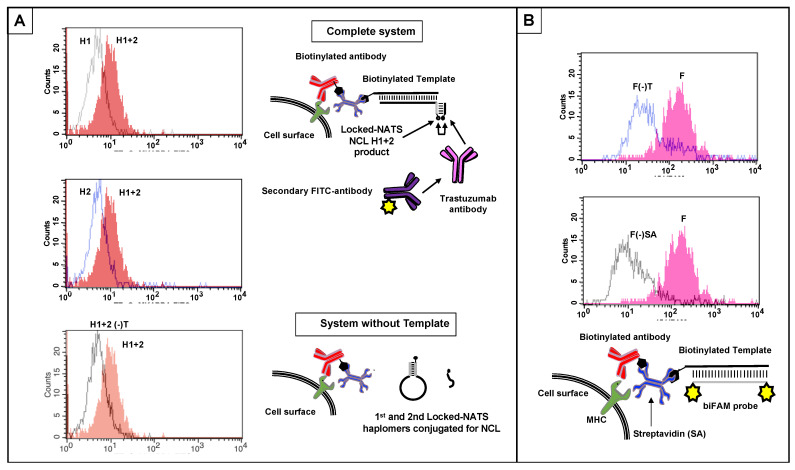
Cell-surface NCL product detection by flow cytometry using Locked-NATS, with templates positioned via anti-MHC Class I antibody and streptavidin, and controls (Methods). (**A**) Locked-NATS haplomers derivatized by click reaction with NCL thioester and cysteinyl mimotope fragments were prepared as for Figure 8. Jurkat cells were prepared with biotinylated surface template (biotin–AGCTGTGTCCTGAAGAAAAGCAAAGACATCTGGACAA) via biotinylated anti-Class I MHC antibody with streptavidin adaptor (Methods; Appendix A). After washing, cells were resuspended at 10^6^ cells/mL in 500 µL PBS, and First and Second NCL haplomers were added to 0.5 µM final. In the presence of template, the First haplomer can anneal and open for subsequent hybridization of the Second haplomer and proximity-driven NCL, as depicted in the top Schematic diagram. In the absence of template, neither First haplomer opening nor cell-surface association can occur (bottom schematic). H1 + H2 = both First and Second haplomers added; H1 = First haplomer alone added; H2 = Second haplomer alone added; H1 + H2(-)T = both First and Second haplomers added to cells bearing all system components except template. Each of the three flow trace panels show the H1 + H2 response separately overlaid on each control for clarity. (**B**) Parallel assessment of presence of Jurkat cell surface template with a biFAM-labeled probe (6Fam- TCCAGATGTCTTTGCTTTTCTTCAGGACACA -6Fam; Methods). F = biFAM probe with complete system; F(-)T = biFAM probe with system bearing all components except template; F(-)SA = biFAM probe with system bearing all components except streptavidin. For clarity, flow trace results are shown in two separate overlay panels to compare controls with complete-system fluorescence.

## Data Availability

All data that support the findings and conclusions of this study are presented in the main text of the paper and in Appendix A, available online.

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
