# Peer review of "Assembly of Biologically Functional Structures by Nucleic Acid Templating: Implementation of a Strategy to Overcome Inhibition by Template Excess"

_molecules, 2022, doi:10.3390/molecules27206831_

Round 1

Reviewer 1 Report

Authors reported chemical ligations using DNA templates.

The whole)

No additional experiments are necessary. However, the manuscript is extremely long and not to the point. I feel that only the essential figures should be kept and that some figures should be erased without moving them to SI. Are Figs. 1,2,5-9 essentially necessary? In addition, kinds of abbreviations should be reduced, because readers will confuse the extremely large number of author-specific words. In addition, with the reduction of figures, the manuscript should be rewritten to be entirely shorter, stating only the essentials.

The term "TAPER" should not be used. "Chemical ligations using tenmplates" should be used, because many researchers have already reported strategies using templates, and the term TAPER does not have novelty. Therefore, using the term "TAPER" gives the impression that the authors have made a new discovery, ignoring the achievements of their predecessors.

The explanation should be complete, including abbreviations, in a single figure. It should not be omitted even if it is explained in the text or in another figure.

1) Panels A and B in Fig. 1 are figures that should be described in Fig. 2 and thereafter, respectively. Panel C does not make sense to readers. Therefore, Fig. 1 should be deleted.

2) Fig. 2 is not the most important figure for the main body of the paper and should be moved to SI.

3) In Fig. 3 panels A and B, authors need to explain what the black squares mean.

Oligo-440 and Peptide-A444 give the impression that they are different compounds, which confuses readers; it should be Oligo-peptide-hybride-1, etc.

Are #444 and A444 in panel C different compounds since they are not described in panels A and B? Also, does "#440 + A444" mean that #440 and A444 reacted?

Is panel D essentially necessary? The difference between each lane is not clear to readers from the figure legends; the words DBCO and TCEP before the abbreviation of DBCO and TCEP are needed in the figure legends separately from the text.

4) If Fig.4 is linked to Fig.3, abbreviations should be used in a unified manner. If not, they should be explained one by one. In Panel A, readers cannot intuitively understand the difference between each lane.

5) Figs. 5-9 are so unintuitive to readers that they seem essentially unnecessary.

6) In Fig. 10, Panel A, authors should add figures like Figs. 3A and B so that readers can intuitively understand *, 1a, and 1b in particular.

DNA sequences should be shown in the figures, not in SI.

7) The intuitive figure in Fig. 11 should be larger. Also, DNA sequences should be shown in the figures, not in SI.

Reviewer 2 Report

Lawler reported in the manuscript (titled “Assembly of Biologically Functional Structures by Nucleic Acid Templating: Implementation of a Strategy to Overcome Inhibition by Template Excess) their efficient TAPER approach of mimotope assembly. Typically, Lk-TAPER is based on the stem-loop structure, which in tandem with template recognition, leads to better than the haplomers. This work adds to the literature for peptide assembly and offer a clear and significant advance over existing methodologies. As is to me the only inadequacy is the quality of the figure shows with poor quality and some garbled text in the figure. 

Round 2

Reviewer 1 Report

The manuscript is a little better, but still difficult for readers to read. If the figures are not written intuitively or are too small in size, readers cannot understand them. There are also many garbled characters in the figures. Based on this point, the manuscript is "Extensive editing of English language and style required" and needs further revision.

Further revisions will raise new questions, so the next revision will not be the end of revising. Therefore, the authors need to make the figures clearer and larger.

Importance: It appears that the authors have created the figures on the assumption that the main text and figure legends should be read and understood. In a chemistry journal such as Molecules, the figures alone should provide the reader with an intuitive understanding of the article as a whole.

Fig.1

Aren't Panels A-C and D directly related? Shouldn't Panels A-C be moved to Fig. 2 or later?

A new panel should be added in Fig. 1 to explain Panel D.

Although the closed square is explained, the structure of the reaction product of DBCO and azide should be written directly in the figure. The reason is that the structure of the NCL reaction is shown. This journal is "Molecules".

When the explanation of Panel D was explained by Panels B and C, the compounds Oligo-441, 440, 439, Peptide-A444, A423 before NCL reaction and Haplomer a, b, c after NCL reaction are mixed and described in the figure. Therefore, readers are confused because of this. Oligo-441, 440, 439, Peptide-A444, A423 should not be shown as reaction products of DBCO and azide.

Fig.2

The difference between Lanes 5-7 should be clearly indicated in the figure.

What do the arrows on lane 6 indicate?

Fig.3

What is "Architecture 2 (Fig1)"?

Panels A-D are too small. The size should be increased.

The meaning of Panel A is not clear. Is it necessary? If it is necessary, it should be changed to a figure more understandable to readers. Or it should be removed. Either way, Panel A is too small to read.

Are the measured datapoints in Panel D the same as in Panel B? If so, it should be mentioned in the figure legends of Panel D that it is the same as Panel B.

Fig.4

Figure is too small. Should be doubled in size.

The difference between a, b, c, d, and e should be clearly indicated in Panel B.

I don't understand the meaning of Panel A. It should be redesigned to be easier to understand.

Fig.5

Is this figure not necessary?

Either way, there is no Panel like Fig. 1A, B, C, Fig. 4A. Without adding such a Panel, readers cannot understand the figure alone.

Fig.6

Same as Fig.5.

Fig.7

The figure is too small and should be doubled in size.

The text is garbled. It is doubtful whether the authors have really checked the manuscript carefully.

Fig.8

The figure is too small and should be doubled in size.

The text is garbled.

The products after NCL reaction should also be shown as in Fig.1A-C.

Fig.9

The figure is too small and should be doubled in size.
